# Value of dynamic clinical and biomarker data for mortality risk prediction in COVID-19: a multicentre retrospective cohort study

Carlo Berzuini,[1] Cathal Hannan,[2] Andrew King,[2] Andy Vail,[1] Claire O'Leary,[3] David Brough,[3] James Galea,[4] Kayode Ogungbenro,[5] Megan Wright,[2] Omar Pathmanaban,[2] Sharon Hulme,[3] Stuart Allan,[3] Luisa Bernardinelli,[6] Hiren C Patel [ID] [2]

For numbered affiliations see end of article.

**Correspondence to**
Mr Hiren C Patel;
hiren.Patel@srft.nhs.uk

## ABSTRACT

**Objectives** Being able to predict which patients with COVID-19 are going to deteriorate is important to help identify patients for clinical and research practice. Clinical prediction models play a critical role in this process, but current models are of limited value because they are typically restricted to baseline predictors and do not always use contemporary statistical methods. We sought to explore the benefits of incorporating dynamic changes in routinely measured biomarkers, non-linear effects and applying 'state-of-the-art' statistical methods in the development of a prognostic model to predict death in hospitalised patients with COVID-19.

**Design** The data were analysed from admissions with COVID-19 to three hospital sites. Exploratory data analysis included a graphical approach to partial correlations. Dynamic biomarkers were considered up to 5 days following admission rather than depending solely on baseline or single time-point data. Marked departures from linear effects of covariates were identified by employing smoothing splines within a generalised additive modelling framework.

**Setting** 3 secondary and tertiary level centres in Greater Manchester, the UK.

**Participants** 392 hospitalised patients with a diagnosis of COVID-19.

**Results** 392 patients with a COVID-19 diagnosis were identified. Area under the receiver operating characteristic curve increased from 0.73 using admission data alone to 0.75 when also considering results of baseline blood samples and to 0.83 when considering dynamic values of routinely collected markers. There was clear non-linearity in the association of age with patient outcome.

**Conclusions** This study shows that clinical prediction models to predict death in hospitalised patients with COVID-19 can be improved by taking into account both non-linear effects in covariates such as age and dynamic changes in values of biomarkers.

## INTRODUCTION

Most patients with SARS-CoV-2 experience mild symptoms. Some patients, however, experience significant symptoms requiring

### Strengths and limitations of this study

► Incorporating routinely available blood tests performed over the first 5 days of hospital admission with clinical presentation data can enhance patient-level prediction of COVID-19 progression.

► A larger dataset is needed to construct definitive prediction models.

► More sophisticated statistical exploitation of biomarker trajectories, for example, using random-effects models of biomarker evolution or 'conditional on outcome' models of biomarker evolution, could make clinical predictions models still better.

hospitalisation. The pandemic nature of the COVID-19 outbreak has meant that hospital services and capacity can be overwhelmed.[1] A tool to predict which patients are likely to deteriorate or need intensive care would help clinicians, hospital managers and researchers make better decisions.

Several such models are reported for patients with COVID-19 but have been criticised for risk of bias using the Prediction model Risk Of Bias ASsessment Tool's criteria.[2] We have further concerns regarding the statistical tools used to develop models. First, current models typically only consider patient characteristics available at baseline and do not consider that presentation of patients with COVID-19 and in hospital course is variable. Second, models routinely seek only linear effects of potential predictors on the outcome of interest, although these are not always clinically plausible.

We sought here to explore the benefits of incorporating dynamic changes in routinely measured biomarkers, non-linear effects and applying 'state-of-the-art' statistical methods

BMJ

**Table 1** Demographic, clinical and medical history factors considered at baseline (ICU intensive care unit; MAP mean arterial pressure)

| | Overall dataset |
|---|---|
| Number of patients | 392 |
| Age, median (IQR) | 71 years (22 years) |
| Gender: male:female ratio | 65:35 |
| Median time to hospitalisation following disease onset (IQR) days | 5 (8) |
| Initial symptoms (%) | |
| Fever | 223 (57) |
| Cough | 240 (61) |
| Dyspnoea | 245 (65) |
| Fatigue | 127 (37) |
| Muscle ache | 53 (16) |
| Comorbidities | |
| Cardiovascular disease | 108 (28%) |
| Chronic respiratory disease (inc asthma) | 110 (28%) |
| Chronic renal disease | 45 (12%) |
| Chronic liver disease | 14 (2%) |
| Obesity | 34 (10%) |
| Diabetes | 95 (24%) |
| Dementia | 49 (13%) |
| Current smoker | 24 (7%) |
| Presenting clinical features | |
| Requirement for supplemental $O_2$ | 125 (37%) |
| Oxygen saturation <90% | 59 (17%) |
| Respiratory rate >24 | 109 (30%) |
| Temperature ≥38°C | 168 (45%) |
| MAP <70 mm Hg | 30 (8%) |
| Outcomes | |
| Acute Respiratory Distress Syndrome | 47 (17%) |
| Non-invasive ventilation | 25 (9%) |
| Need for ICU care | 31 (12%) |
| Invasive ventilation | 14 (5%) |
| Death | 110 (27%) |

in the development of a prognostic model to predict death in hospitalised patients with COVID-19.

## METHODS
### Study population
Admissions with confirmed COVID-19 (according to WHO guidance) at three hospitals in the Northern Care Alliance (Greater Manchester, the UK) between 11 March and 17 April 2020 with a minimum of a 3-week follow-up were studied.[3]

### Data collection
Necessary approvals were obtained from the local research and innovation department. Research nurses abstracted data from the electronic patient records based on the International Severe Acute Respiratory and emerging Infection Consortium (ISARIC) data collection tool but modified for use with this study.[4] The ISARIC study data were supplemented from electronic patient records with results of blood analyses performed as part of routine clinical care. The date of diagnosis was considered day 1.

### Data analysis
All data were subjected to range checks and validated for internal consistency and missing items then anonymised prior to transfer.

### Selection of biomarkers
The initial list of potential markers was determined through review of the literature and availability within routinely collected data. Candidate variables were further screened using a graphical representation of the partial correlation structure stratified by survival status.[5] Routine bloods were typically analysed on alternate days. The assumption that the unrecorded values were missing at random was corroborated by inspection of joint bivariate plots of complete and incomplete observations made on each particular marker on consecutive days.[6] Then, each missing value of a marker was imputed by iterative sampling from its conditional predictive distribution given past values, using R's MICE package.[6 7]

### Modelling
In this study, we used the information contained in the clinical presentation data and available biomarkers (creatinine, lymphocyte count, etc) to update, on a day-by-day basis, the patient's probability of death within 21 days.

Initially, a logistic model for the all-cause mortality outcome using only clinical features at presentation was fitted initially. We then fitted separate logistic models for death for each day, using predictive variables identified from the partial correlation analysis described above. For each of the 5 days following hospital admission, we fitted a model based exclusively on data from subjects still alive at that day, with candidate predictors chosen out of the set of clinical variables and biomarker values collected until that day. This approach meant that for each of the first 5 days following admission, a sequence of day-specific mortality prediction models was available. We subsequently fitted each model within a generalised additive modelling framework involving smoothing splines to detect marked departures from linearity for continuous predictors and undertook data transformations (eg, log transformation of concentrations) as indicated.[8] A standard logistic version of the model was then fitted. We used the Akaike Information Criterion to choose between logistic models and assessed predictive performance using the area under a 10-fold, cross-validated receiver operating characteristic (ROC) curve.

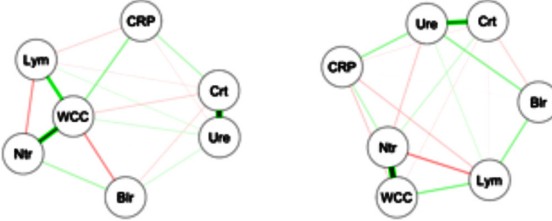

**Figure 1** Partial correlations between biomarkers. Nodes represent average marker levels from day 2 to day5 and edges represent partial correlations, as calculated from the survivors (left) and from the decedents (right). Broader lines indicate stronger relationships. Blr, bilirubin, CRP, C reactive protein, Crt, creatinine, Lym, lymphocytes, Ntr, neutrophils, WCC, white cell count, Ure, urea.

## Patient and public involvement

There was no involvement of patients or the general public in the design or delivery of this study. This was because of the acute nature and fast moving pace of the disease studies and because access to patients and the public was limited at this time.

## RESULTS

A total of 392 patients with a COVID-19 diagnosis were admitted during the study period. Table 1 provides a summary of their demographic and clinical features, including medical history. Blood samples were typically requested every other day following admission (online supplemental table).

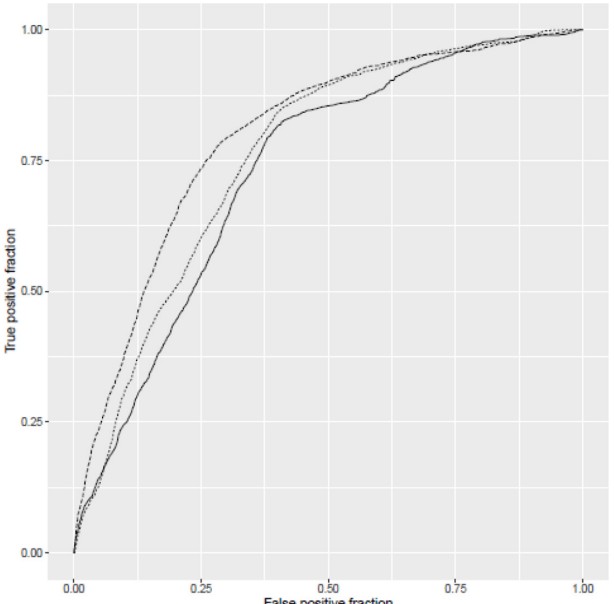

**Figure 2** Receiver operating characteristic (ROC) curves for three models: solid line indicates model considering only clinical factors at baseline (area under ROC curve=0.73); finely dotted line indicates model extended to consider also biomarker data from baseline sample (area under ROC curve=0.75) and top line indicates model at 5 days extended to consider dynamic changes in biomarker data (area under ROC curve=0.83). Note that models are not nested.

For an informal analysis of biomarker relationships, we analysed partial correlations between seven potential inflammatory markers in each of the two survival groups (figure 1). Among survivors, the anticipated correlations between lymphocytes and neutrophils (via white cell count) and between creatinine and urea were present, but these correlations were found to be significantly lower among the decedents, suggesting the possible presence of differences in the neutrophil/lymphocyte and urea/creatinine ratios between the two outcome groups.

Inclusion of admission biomarker data did not improve the predictive value of the model over clinical data alone. Incorporation of post-admission dynamic biomarker data did, however, increase the discriminative ability of the model (figure 2). Estimates from the best fitting model at day 5 (table 2) show strongly statistically significant term(s) reflecting post-baseline biomarker changes that can be readily visualised (figure 3). In addition to age and disease severity, the most recent neutrophil/lymphocyte ratio and the two most recent (and, therefore, recent change in) urea/creatinine ratios were generally predictive. There was a marked non-linearity in the effect of age (figure 4).

## DISCUSSION

These results suggest that using dynamic data is better than using baseline initial presentation data to predict death in patients with COVID-19. Even with a local dataset of just 392 admissions with COVID-19, we were able to identify clear benefit from exploiting dynamic biomarker data and marked non-linearity in the effects of commonly used factors to predict outcomes. Our findings should be taken as indicative of the benefits of 'state-of-the-art' statistical methodology but also the necessary collaboration between statisticians and clinicians as this statistical methodology is not readily accessible to most researchers. Identification and validation of anything approaching a definitive predictive model would require substantially larger sample sizes.[2]

Neither the non-linear effect of age after allowance for other factors nor the particular biomarkers identified within this dataset are surprising. Others have also observed associations of mortality with age, and clinical and biochemical markers of disease severity (eg, neutrophil/lymphocyte ratio).[9–11] There have been few studies investigating dynamic changes in patient biomarkers for mortality prediction in COVID-19; one such study of 548 patients in China also demonstrated that the neutrophil:lymphocyte ratio in survivors and non-survivors became increasingly divergent throughout their hospital admission.[12] Chen *et al* derived their prognostic score from an analysis based on a Cox's regression model with their candidate predictive variables taken at baseline. They incorporated in their analysis the slope of a line fitted to the first and last measurements of each particular marker to model changes over time. Chen *et al* approach has

**Table 2** Estimated coefficients (Est) with their SE and p value

| Predictor | Clinical data alone: day 1 | | | Clinical data+day 1*, biomarker data | | | Day 3 | | | Day 4 | | | Day 5 | | |
|---|---|---|---|---|---|---|---|---|---|---|---|---|---|---|---|
| | Est | SE | P value | Est | SE | P value | Est | SE | P value | Est | SE | P value | Est | SE | P value |
| Intercept | -5.37 | 1.46 | 0.0003 | -4.36 | 1.35 | 0.001 | 0.67 | 1.97 | 0.73 | 0.005 | 1.86 | 0.99 | -0.20 | 1.68 | 0.9 |
| Log Neut/Lymp D1 | | | | 0.28 | 0.16 | 0.08 | | | | | | | | | |
| Log Neut/Lymp D3 | | | | | | | 0.41 | 0.19 | 0.03 | | | | | | |
| Log Neut/Lymp D4 | | | | | | | | | | 0.48 | 0.2 | 0.02 | | | |
| Log Neut/Lymp D5 | | | | | | | | | | | | | 0.52 | 0.21 | 0.01 |
| Log Urea/Creat D2 | | | | | | | -4.22 | 1.24 | 0.0007 | | | | | | |
| Log Urea/Creat D3 | | | | | | | 5.13 | 1.30 | 0.0001 | | | | | | |
| Log Urea/Creat D4 | | | | | | | | | | 1.08 | 0.35 | 0.002 | -4.97 | 1.72 | 0.0003 |
| Log Urea/Creat D5 | | | | | | | | | | | | | 6.32 | 1.77 | 0.0004 |
| Age (years) | 0.064 | 0.012 | <0.0001 | 0.073 | 0.012 | <0.0001 | 0.069 | 0.013 | <0.0001 | 0.071 | 0.012 | <0.0001 | 0.066 | 0.012 | <0.0001 |
| O₂ saturation | -0.028 | 0.012 | 0.01 | -0.03 | 0.012 | 0.01 | -0.03 | 0.013 | 0.03 | -0.03 | 0.012 | 0.02 | | | |
| Respiratory rate | 0.085 | 0.022 | 0.0001 | 0.085 | 0.022 | 0.0001 | 0.08 | 0.023 | 0.0003 | 0.087 | 0.022 | 0.0001 | 0.09 | 0.022 | 0.0001 |
| Smoking | 0.7 | 0.263 | 0.0073 | 0.7 | 0.267 | 0.01 | 0.8 | 0.27 | 0.004 | 0.71 | 0.27 | 0.008 | 0.76 | 0.28 | 0.006 |

Note that different variables are selected at different days so that models are not nested.

* The addition of biomarker data on day 2 did not contribute any additional predictive power of that obtained on day 1.

Creat, creatinine; D, day; Neut/Lymp, neurotrophil/lymphocyte; O₂, oxygen.

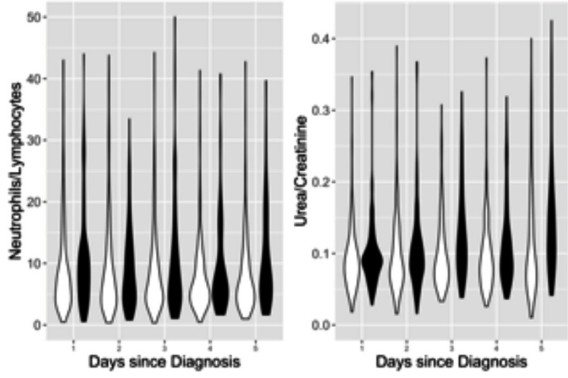

**Figure 3** Violin plots showing distribution at each day of admission, stratified by survival status, for biomarkers identified by statistical modelling. Panel A: log-transformed neutrophil ($\times 10^9$/L)/lymphocyte ($\times 10^9$/L) ratio. Panel B: log-transformed urea (mmol/L)/creatinine (µmol/L) ratio. Survivors (white) on left and decedents (shaded) on right.

advantages and disadvantages. Their model captures duration information but does not allow choice of time horizon for prediction. Their predictions are arguably limited because they are not updated daily and depend on the assumption that marker evolution is linear and summarised by a straight line between initial and final values.

A smaller study limited to patients with severe COVID-19 also revealed a progressive increase in neutrophil count and plasma interleukin-6 concentration in the decedents when compared with the survivors, but the authors did not perform any assessment of the predictive value associated with dynamic changes in these laboratory parameters.[13]

Similarly, renal injury has also been shown to be common in patients with COVID-19 and is associated

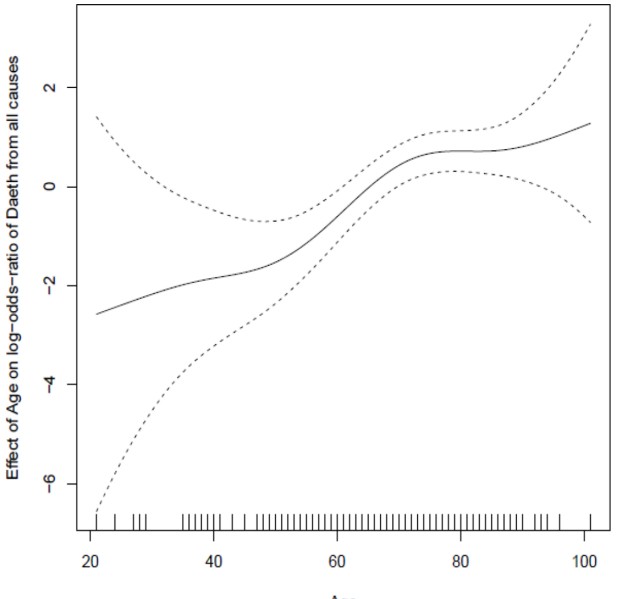

**Figure 4** Spline plot demonstrating marked non-linearity in relationship between age and outcome after adjustment for other factors included in the final model.

with a worse outcome.[14] The reason for this is not clear. There is emerging evidence that SARS-CoV-2 infection can directly harm the kidneys. The worsening urea/creatinine ratio observed in our data set may also reflect either the therapeutic effects of fluid restriction to treat severe Acute Respiratory Distress Syndrome (ARDS) or evidence of multi-organ dysfunction.[10 15] Regardless of the cause, the impact of the urea/creatinine level on death was not evident at presentation but became a significant predictor of death in our model over time. This observation illustrates the benefit of taking into account improvements and deterioration in daily blood test results as well as initial presentation factors when calculating the probability of death. Improving the accuracy of prediction models using this approach is likely to be successful in informing clinical decision-making, resource planning and communication with patients and relatives.

We acknowledge that our study has some limitations. We would have liked to consider other outcomes, including dynamic changes in clinical variables, as well as disease endpoints such as the incidence of ARDS and ICU (intensive care admission) admission. Dynamic clinical data were not included because it was less reliable to obtain compared with blood biomarker data, and a consistent diagnosis of ARDS, and dates of onset or admission to ICU were also not routinely available. Although we have undertaken internal cross-validation to ensure unbiased comparison of ROC curves, we have not considered calibration. We do not wish to make any claim for the value of our current models at each day based on the small sample size available to us locally. With only three hospital sites contributing during the first wave, and because of significant time/resource pressures during the pandemic, we did not have sufficient data to construct definitive prediction models or to follow-up patients beyond 3 weeks. More sophisticated exploitation of biomarker trajectories through, for example, approaches based on random-effects models of biomarker evolution or 'conditional on outcome' models of biomarker evolution, would also require more data and be expected to add further insights.[16–18]

Clinical prediction models are important and can help in clinical decision-making, resource allocation and optimal selection of trial participants for investigational treatments. In the setting of an infectious disease pandemic—affecting all geographic and socio-economic groups—using routinely available blood tests to inform prediction models has obvious advantages over less widely available, but perhaps more specific, biomarkers of disease severity. Until investigators incorporate such data in participant selection, it is unlikely that future trials will be able to accurately target those patients most likely to benefit from therapies such as immunomodulation. Overall benefits will be 'diluted' and potentially reversed by inclusion of participants who have nothing to gain and,

in theory, may be harmed by restriction of a healthy inflammatory response.[19] The consequence of poorly considered eligibility criteria may, therefore, be to erroneously dismiss therapies that could benefit those at highest risk from COVID-19.

**Author affiliations**
[1]Centre for Biostatistics, The University of Manchester, Manchester Academic Health Sciences Centre, Manchester, UK
[2]Manchester Centre for Clinical Neurosciences, Salford Royal Hospitals NHS Trust, Salford, UK
[3]Division of Neuroscience and Experimental Psychology, The University of Manchester, Manchester, UK
[4]Cardiff and Vale University Health Board, Cardiff, UK
[5]Department of Pharmacy and Optometry, The University of Manchester, Manchester, UK
[6]Department of Brain and Behavioural Sciences, The University of Pavia, Pavia, Italy

**Correction notice** This article has been corrected since it first published. Table 2 has been updated.

**Acknowledgements** The Northern Care Alliance research delivery team who collected the data for the patients, and the Salford Royal business analysis team who helped in providing access to the daily blood and clinical observation results. This report was also made possible by the provision of the data collection tool through the efforts and expertise of the International Severe Acute Respiratory and emerging Infection Consortium's Team (https://isaric.tghn.org/).

**Contributors** All authors (CB, CH, AK, AV, CO'L, DB, JG, KO, MW, OP, SH, SA, LB and HCP) were involved in the concept and design of the study, which was led by HCP and AV, and involved in revising the manuscript and contributed to the final draft. HCP, CH and MW acquired the data, and CB, LB, AV, HCP and CH analysed and interpreted the data. CB, DB, SA, OP, JG, AV, CH and HCP drafted the manuscript. The corresponding author attests that all listed authors meet authorship criteria and that no others meeting the criteria have been omitted. HCP acts as guarantor for the study.

**Funding** AK, AV, JG, HCP and SH are funded by the National Institute for Health Research Efficacy and Mechanism Evaluation Programme, Ref: 14/209/07. DB is funded by MRC grant MR/T016515/1.

**Competing interests** Swedish Orphan Biovitrum have provided investigational medicinal product for public-funded, peer-reviewed trials on which AK, AV, JG, HCP and SH are coinvestigators. The other authors declare no competing interests.

**Patient and public involvement** Patients and/or the public were involved in the design, or conduct, or reporting, or dissemination plans of this research. Refer to the Methods section for further details.

**Patient consent for publication** Not required.

**Provenance and peer review** Not commissioned; externally peer reviewed.

**Data availability statement** Data are available upon reasonable request.

**ORCID iD**
Hiren C Patel http://orcid.org/0000-0002-1439-8801

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
