## [Reviewer comments · BMJ Open]

ARTICLE DETAILS

TITLE (PROVISIONAL)	The value of dynamic clinical and biomarker data for mortality risk prediction in COVID-19: A multi-centre retrospective cohort study
AUTHORS	Berzuini, Carlo; Hannan, Cathal; King, Andrew; Vail, Andy; O'Leary, Claire; Brough, David; Galea, James; Ogungbenro, Kayode; Wright, Megan; Pathmanaban, Omar; Hulme, Sharon; Allan, Stuart; Bernardinelli, Luisa; Patel, Hiren

VERSION 1 – REVIEW

REVIEWER	John Frater Washington University USA
REVIEW RETURNED	07-Jul-2020

GENERAL COMMENTS	This is a review of "Dynamic clinical and biomarker data for mortality risk prediction in COVID-19" by Berzuini et al. This is a study from Britain reviewing the data for a cohort of 392 hospitalized with COVID-19 infection. In contrast to earlier studies, the current study analyzed multiple time points rather than admission biomarker data. I have the following specific comments. 1. The quality of the English is appropriate for a scientific study. However, it is unclear to me who the intended audience for this manuscript is. I suspect that the way the work is presented, the authors are primarily communicating with researchers interested in statistical modelling of COVID-19 infection. If they are interested in having a readership that includes clinicians, they should modify their manuscript to make their statistical work more accessible to that audience. The authors write in their discussion: "Our findings should be taken as indicative of the benefit of applying more recent developments in statistical methodology than are commonly found in the clinical literature". It would be useful to explain these techniques in greater detail so they can reach a broader audience.2. It would be worthwhile for the authors to comment on other studies that looked at longitudinal biomarker data, rather than single time point.
---

REVIEWER	Matthieu Jabaudon CHU Clermont-Ferrand and Université Clermont Auvergne, France. Vanderbilt University Medical Center, USA.
REVIEW RETURNED	12-Jul-2020

GENERAL COMMENTS	This is an interesting study of a prediction model for all-cause mortality based on baseline and longitudinal clinical and biological
---

	data analysis from 392 patients with Covid-19 admitted to three UK hospital sites. The originality and the main message of the study are to emphasize that the evolution in patient characteristics should be considered to build such a prediction model, not only their measurements at baseline. In this perspective, the manuscript is very well-written, analyses are very sound, and based on original and robust methods, and the findings are very valuable. However, and as reported by the authors themselves, such findings should be mostly considered as hypothesis-generating rather than they bring definitive conclusions, especially because of the “limited” sample size; indeed, although the investigators should be commended for enrolling 392 patients in their study, which is a lot, of course, this number may be too small to draw robust conclusions, which in my opinion does not jeopardize the interest of the current manuscript. Major comments  - It is unclear to me why the authors did not also consider an approach to the dynamics of clinical variables (such as those used at baseline) the same way they did for biological data. I guess the “clinical” course (i.e. the evolution of clinical variables over the 5 first days after diagnosis, e.g. the development of shock, AKI, etc.) would be very informative too but, unless I misunderstood, it was not considered. Why? Could the authors explain and maybe add this point as a limitation here? - As said above, I wonder whether the authors might have been able to anticipate that they would not reach sufficient power to construct definitive prediction models, based on the number of patients included in the analysis. Although this does not lessen the importance of their hypothesis-generating findings, I would suggest adding some explanations on the overall context of the study to make it easier to understand by the readers (limited time for enrolling more patients, overwhelmed centers because of Covid-19, etc.) - Why was a 3-week duration (compared to 28 days for example) decided for the follow-up? This choice could benefit from some further explanations. Minor comments  - An interesting finding of the study is the impact of urea/creatinine level on the risk of death. Would it be possible to evaluate in a more broadly manner the presence of AKI per se (such as through the use of current scores such as the KDIGO) or the need for CRRT as predictors, to reinforce these findings? - In the same perspective, have the authors considered analyzing some severity scores routinely used in the ICU (such as SOFA or APACHE) as they would combine most of the (clinical and biological) variables that were analyzed separately here? - Although this is true, I would not repeat twice the terms “state of the art” in the “objectives” section of the abstract.
--	--

VERSION 1 – AUTHOR RESPONSE

Reviewer: 1

We have included greater detail in the methods section on modelling to allow for a better

understanding of what was done to try and reach a broader audience.

'In this study we used the information contained in the clinical presentation data and available biomarkers (creatinine, lymphocyte count, etc.) to update, on a day-by-day basis, the patient's probability of death within 21 days

Initially, a binary logistic model for the all-cause mortality outcome using only clinical features at presentation was fitted initially. We then fitting separate logistic models for death for each day, using predictive variables identified from the partial correlation analysis described above. For each of the five days following hospital admission, we fitted a model based exclusively on data from subjects still alive at that day, with candidate predictors chosen out of the set of clinical variables and biomarker values collected until that day. This approach meant that for each of the first five days following admission, a sequence of day specific mortality prediction models were available. We subsequently fitted each model within a generalised additive modelling (GAM) framework involving smoothing splines to detect marked departures from linearity for continuous predictors and undertook data transformations (e.g. log transformation of concentrations) as indicated.⁸ A standard logistic version of the model was then fitted. We used the Akaike Information Criterion (AIC) to choose between logistic models and assessed predictive performance using the area under a ten-fold, cross-validated Receiver Operating Characteristic (ROC) curve'.

We have also emphasised in the discussion that we have used state of the art statistical methodology that is not available to most clinicians and outlined that that there needs to be a meaningful collaboration between statistics and clinicians to get the most out of this technology.

'Our findings should be taken as indicative of the benefits of 'state of the art' statistical methodology but also the necessary collaboration between statisticians and clinicians as this statistical methodology is not readily accessible to most researchers'

2. It would be worthwhile for the authors to comment on other studies that looked a longitudinal biomarker data, rather than single time point.

The studies which have used longitudinal data have been added in and we have discussed the approaches used by Chen et al and compared them to our approach.

'There have been few studies investigating dynamic changes in patient biomarkers for mortality prediction in COVID-19; one such study of 548 patients in China also demonstrated that the neutrophil:lymphocyte ratio in survivors and non-survivors became increasingly divergent throughout their hospital admission.¹³ Chen et al derived their prognostic score from an analysis based on a Cox's regression model with their candidate predictive variables taken at baseline. They incorporated in their analysis the slope of a line fitted to the first and last measurements of each particular marker to model changes over time. Chen et al approach has advantages and disadvantages. Their model captures duration information but does not involve choice of time horizon for prediction. Their predictions are arguably limited because they are not updated daily and depend on the assumption that marker evolution is linear and summarised by a straight line between initial and final values. A smaller study limited to patients with severe COVID-19 also revealed a progressive increase in neutrophil count and plasma interleukin-6 concentration in the decedents when compared to the survivors, but the authors did not perform any assessment of the predictive value associated with dynamic changes in these laboratory parameters.¹⁴ '.

Reviewer: 2

We have added in other limitations in our data in the section pertaining to this in the discussion. We would have liked to consider other outcomes including dynamic changes in clinical variables, as well

as disease end points such as ARDS and ICU admission. Clinical data was not included because it was less reliable to obtain compared to blood biomarker data, and a consistent diagnosis of ARDS, and dates of onset or admission to ICU were also not routinely available. Although we have undertaken internal cross-validation to ensure unbiased comparison of ROC curves we have not considered calibration. We do not wish to make any claim for the value of our current models at each day based on the small sample size available to us locally. With only three hospital sites contributing during the first wave, and because of time/resource pressures during the pandemic we did not have sufficient data to construct definitive prediction models or to follow-up patients beyond 3 weeks.

Minor comments

This was a hospitalised group of patients rather than those going to ICU alone and therefore we did not feel that these severity scores were applicable to our population. We also did not have the granularity of data that would be required to construct these disease severity score.

“state of the art” in the “objectives” section of the abstract has been addressed.

I hope that this answers the very helpful comments provided by the reviewers.

VERSION 2 – REVIEW

REVIEWER	John L Frater Washington University USA
REVIEW RETURNED	08-Aug-2020

GENERAL COMMENTS	Thank you. My concerns have been addressed.
---

REVIEWER	Matthieu Jabaudon CHU Clermont-Ferrand, Université Clermont Auvergne, France and Vanderbilt University Medical Center, USA
REVIEW RETURNED	05-Aug-2020

GENERAL COMMENTS	The authors have made important revisions and answered all points raised by the reviewers; I do not have further comments or questions at this stage.
---